# Automatic Disease Detection of Basal Stem Rot Using Deep Learning and Hyperspectral Imaging

**Lai Zhi Yong** [1], **Siti Khairunniza-Bejo** [1,2,3,*], **Mahirah Jahari** [1,2] and **Farrah Melissa Muharam** [4]

1. Department of Biological and Agricultural Engineering, Faculty of Engineering, Universiti Putra Malaysia (UPM), Serdang 43400, Selangor, Malaysia
2. Smart Farming Technology Research Centre, Universiti Putra Malaysia (UPM), Serdang 43400, Selangor, Malaysia
3. Institute of Plantation Studies, Universiti Putra Malaysia (UPM), Serdang 43400, Selangor, Malaysia
4. Department of Agriculture Technology, Faculty of Agriculture, Universiti Putra Malaysia (UPM), Serdang 43400, Selangor, Malaysia
* Correspondence: skbejo@upm.edu.my

**Abstract:** Basal Stem Rot (BSR), a disease caused by *Ganoderma boninense* (*G. boninense*), has posed a significant concern for the oil palm industry, particularly in Southeast Asia, as it has the potential to cause substantial economic losses. The breeding programme is currently searching for *G. boninense*-resistant planting materials, which has necessitated intense manual screening in the nursery to track the progression of disease development in response to different treatments. The combination of hyperspectral image and machine learning approaches has a high detection potential for BSR. However, manual feature selection is still required to construct a detection model. Therefore, the objective of this study is to establish an automatic BSR detection at the seedling stage using a pre-trained deep learning model and hyperspectral images. The aerial view image of an oil palm seedling is divided into three regions in order to determine if there is any substantial spectral change across leaf positions. To investigate if the background images affect the performance of the detection, segmented images of the plant seedling have been automatically generated using a Mask Region-based Convolutional Neural Network (RCNN). Consequently, three models are utilised to detect BSR: a convolutional neural network that is 16 layers deep (VGG16) model trained on a segmented image; and VGG16 and Mask RCNN models both trained on the original images. The results indicate that the VGG16 model trained with the original images at 938 nm wavelength performed the best in terms of accuracy (91.93%), precision (94.32%), recall (89.26%), and F1 score (91.72%). This method revealed that users may detect BSR automatically without having to manually extract image attributes before detection.

**Keywords:** automatic disease detection; *Ganoderma boninense*; hyperspectral imaging; deep learning

## 1. Introduction

*Ganoderma boninense (G. boninense)* which can cause Basal Stem Rot (BSR) disease has been threatening the oil palm plantations in Southeast Asia for decades and it can cause up to USD 500 million loss annually [1–3]. The young palm trees usually die within 2 years after the first shown symptoms such as yellowing and necrotic leaves, small canopy, stunted growth and unopen spear [4,5]. Proper management and monitoring of the oil palm plantation can help control BSR. One of the approaches is to introduce *Ganoderma* spp. resistant planting materials. The breeding of *Ganoderma* spp. resistant planting materials not only reduces the economic impact of the yield loss, but also helps to create a more sustainable production [6–8]. According to Turnbull et al. [8], if young plants are planted too close to infected palms from the previous generation, the infection might occur much earlier, thus leading to plantation expansion. By introducing *Ganoderma* spp. resistant planting materials, the need for expansion might be avoided, which can help preserve the forest, reducing its environmental impact.

For breeding purposes in a nursery, the worker has to screen the seedlings to differentiate BSR-infected and uninfected seedlings manually. Commonly, for confirmation of the infection, laboratory-based test such as the polymerase chain reaction (PCR), immunofluorescence (IF), fluorescence in situ hybridisation (FISH), enzyme-linked immunosorbent assay (ELISA), flow cytometry (FCM) and gas chromatography–mass spectrometry (GC-MS) may be used, which involves destruction of the samples [9]. Several sensing methods have been developed for non-destructive detection of BSR, including methods involving tomography [10–12], e-nose [13,14], spectroscopy [15–17], thermal imaging [18,19], hyperspectral imaging [1,20–23], lidar [24], terrestrial laser scanning [25], and soil sensing [26].

Li et al. [27] has undertaken a thorough review of various types of plant disease detection using deep learning, and concluded that deep learning is capable of identifying plant leaf diseases with high accuracy. Together with hyperspectral imaging, an early detection of plant disease may be obtained. Furthermore, transfer learning and hyperspectral imaging may be used in rice disease detection [28]. A self-designed CNN model was trained to classify disease with one variety of rice, and the learning was transferred to another three varieties of rice. Fine-tuning, deep Correlation Alignment (CORAL), and deep domain confusion (DDC) were three deep transfer learning approaches that were applied, and they each produced accuracy results of up to 93.33%, 86.67%, and 83.33%, respectively. Although there is currently little research performed on hyperspectral imaging in deep learning, the method has been widely applied to RGB images. Su et al. [29] used Mask RCNN for Fusarium head blight in wheat and it has been shown that the Mask RCNN has high potential for disease detection, achieving precision, recall, F1 score and detection rate at 72.10%, 76.16%, 74.04% and 98.81%, respectively. Feasibility of other state-of-the-art methods such as Faster RCNN, you only look once version 4 (YOLOv4), CenterNet, DetectoRS, Cascade RCNN, Foveabox and Deformabe Detr on the detection of diseased citrus were also studied, where it was indicated that the deep learning-based technique showed good performance in the detection of early stage citrus leaf diseases with CenterNet having the highest accuracy. YOLOv4 had the fastest detection among the models studied [30]. The single shot detector (SSD) was also studied for anthracnose infection on walnut tress where it achieved 87% accuracy on the validation dataset [31]. Aside from all the object detection techniques mentioned, the Mask RCNN was the only method that masked the object rather than only creating bounding boxes around the object, allowing it to be used for image segmentation. Other than Mask RCNN, research studies had been undertaken for VGG16 in plant disease detection. VGG-based transfer learning was able to achieve high average accuracy for cucumber leaf images for seven viral diseases at 93.6% [32]. According to Rahman et al. [33], a fine-tuned VGG16 model achieved a 97.12% accuracy in classifying six classes of pest and disease from rice leaf images. VGG16 was also studied for several disease classifications in tomato plants, where it achieved a net accuracy of 97.23% for seven classes of the diseases studied [34].

Based on the literature review, it can be concluded that hyperspectral imaging has the capacity to detect BSR in the Near-infrared (NIR) region. However, the application of deep learning was not thoroughly studied for BSR detection. Furthermore, no research has been conducted on plant disease identification utilising a pre-trained and widely available deep learning model in the Tensorflow model zoo in conjunction with hyperspectral imagery.

Consequently, the objective of this study is to develop a deep learning model for BSR detection in oil palm plant seedlings that does not rely on human feature extractions using pre-trained models from the Tensorflow model zoo. It contains an analysis of the effect of leaf geometry on wavelength reflectance, an analysis of the effect of image segmentation on model performance, and the identification of the most appropriate model for BSR detection.

## 2. Materials and Methods

As shown in Figure 1, the overall flowchart of the study includes data pre-processing, wavelength selection for BSR detection and background removal, image generation, and augmentation, as well as model development and comparisons. All the analyses in this

study were undertaken on a machine with Intel core i7th generation CPU with an NVIDIA GeForce RTX 2070.

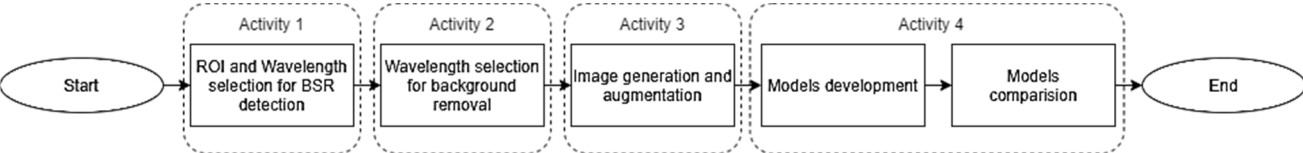

**Figure 1.** Flowchart of the process involved in this study.

### 2.1. Dataset

This study used the same hyperspectral images as the study completed in [22], where the images of 10-month-old oil palm seedlings were captured inside a glasshouse from 11.00 am to 2.00 pm on a sunny day using a Cubert FireflEYE S185 (Cubert Gmbh, Ulm, Germany) snapshot hyperspectral camera at 2.6 m constant height from the ground with a black background. All the oil palm seedlings were subjected to the same treatment, in a controlled environment with constant temperature and humidity. Therefore, the difference in the sample reflectance was assumed to be due to the BSR infection. Sixteen bands that demonstrated great performance regardless of the frond number during BSR disease detection using machine learning techniques in [22] were used in this study, i.e., 890 nm, 894 nm, 898 nm, 902 nm, 906 nm, 910 nm, 914 nm, 918 nm, 922 nm, 926 nm, 930 nm, 934 nm, 938 nm, 942 nm, 946 nm, and 950 nm. These bands were extracted from the bands that have shown great separation value between the infected and uninfected plant seedlings. Further, the bands were also tested for any significant difference between the infected and uninfected plant seedlings using a t-test in SPSS statistical software (IBM SPSS Statistics 25, IBM, New York, NY, USA) in which the p-values of the tests were less than 0.05 and obtained good coefficients of variance that were between 5 and 14%.

### 2.2. Region of Interest (RoI) and Wavelength Selection for BSR Detection

The disease slowly destroys the vascular system and causes symptoms of water and nutrient deficiency. Leaf characteristics display different changing trends under nutritional stress [35]. Therefore, the question if the location of points on the fronds provides significant differences to spectral reflectance, which may thus influence the plant status at early infection, was first investigated. In addition, according to [36], a reflectance spectra is very sensitive to the geometry of the plant. Therefore, to check if there is any effect of the point reflectance at different positions of the leaf structure, an aerial view of an oil palm seedling was divided into three RoIs as shown in Figure 2 and defined as follows:

A:	Inner region—2 cm from the centre of the seedling to 5 cm square.
B:	Middle region—5 cm from the centre of the seedling to 8 cm square.
C:	Outer region—8 cm from centre of the seedling to 11 cm square.

A total of 693 points of reflectance were extracted randomly from 72 hyperspectral images in the *.cub file format using Cube-Pilot software (Cube-Pilot 1.5.8, Cubert Gmbh, Ulm, Germany), where 36 images were obtained from healthy seedlings and 36 images were obtained from infected seedlings. A box plot method was used to remove points that exceeded the lower and upper fences. As a result, only 668 points were left.

Analysis of Variance (ANOVA) was used to check the significant difference between the data groups. However, ANOVA assumes normality. Therefore, before conducting an ANOVA, the modified Wilk–Shapiro test that allows a sample size of more than 30 was used to check for normality [37–40]. First, the range of data was adjusted based on $\mu \pm s$, where $\mu$ is the mean, s is the upper and lower limit testing with different values, i.e., s starting from $2\sigma$ until $0.5\sigma$ with 0.5 decrement where $\sigma$ is the standard deviation. If there were no wavelengths showing normality until $s = 0.5\sigma$, the data went through the second stage which involved data transformation utilising log and reflection transformation. A summary of the process of the normality check and outlier removal is shown in Figure 3.

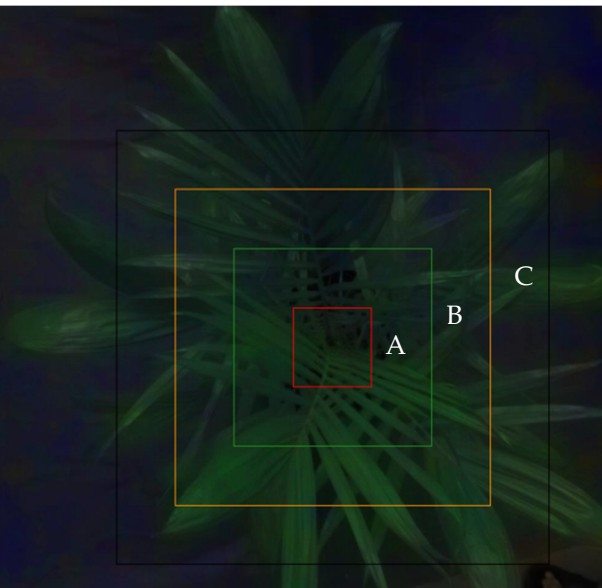

**Figure 2.** The three regions of interest.

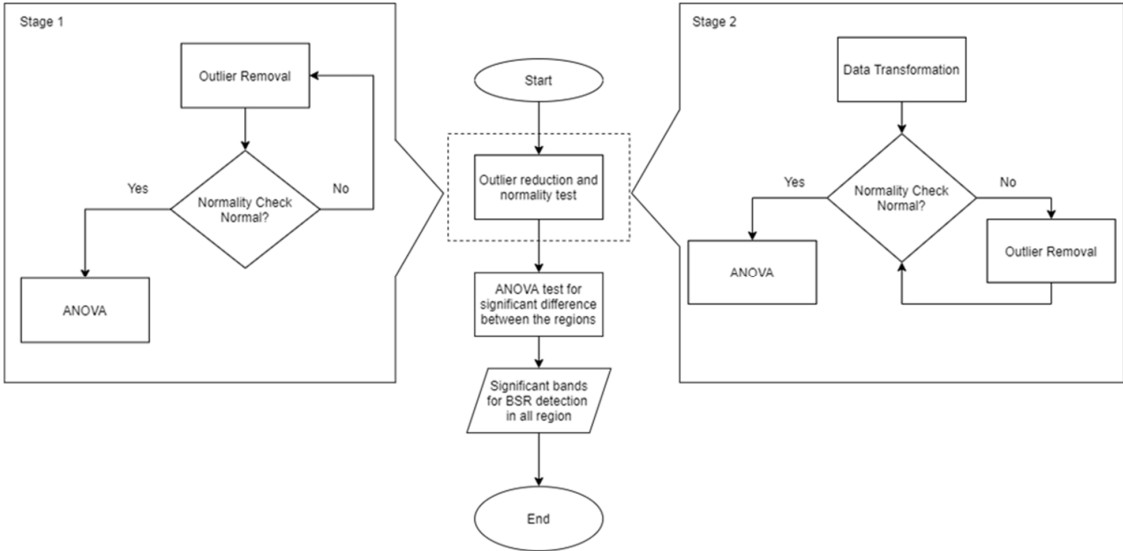

**Figure 3.** Flowchart of normality check and outlier removal.

After the selection of suitable wavelength(s), the wavelength(s) was (were) used to generate images with 1000 by 1000 pixels as input for the model for BSR detection. The details of image generation are discussed in Section 2.4.1.

*2.3. Wavelength Selection for Background Removal*

As mentioned earlier, the effect of segmentation on the model performance was studied. Therefore, a suitable wavelength for image segmentation to generate clear images for background removal was determined based on the difference in reflectance between the plant seedling and the background. The separation was calculated as in Equation (1), where $\overline{Ref_{plant}}$ is the average reflectance of the points and $\overline{Ref_{background}}$ is the average reflectance of the background at wavelength $i$.

$$Separation_i = \left| \overline{Ref_{plant,i}} - \overline{Ref_{background,i}} \right|. \tag{1}$$

The selected wavelengths were used to generate images with 1000 by 1000 pixels for segmentation purposes and the details of image generation are discussed in Section 2.4.2.

### 2.4. Image Generation and Augmentation

After the wavelengths were selected for both BSR detection and background removal, two sets of images were generated. Each set of images was used for different purposes, and the images were fed to different models. The details are discussed in the model development in Section 2.5.

#### 2.4.1. Image Generation for BSR Detection

As discussed in Section 2.2, the ANOVA was used to identify the significant differences between each RoI. This was done to identify which wavelength had a more consistent performance regardless of the geometry of the leaves. The wavelength(s) which had no significant difference between each RoI was (were) chosen. The images were generated by extracting the chosen wavelength from the hyperspectral images using Python. The images were then augmented using rotation, zoom, horizontal and vertical flip. Each set of images was augmented to a total of 1610 images, where 1127 were used for training and 483 for testing. In this study, these images were defined as original images.

#### 2.4.2. Image Generation for Background Removal

The images generated in Section 2.3 were not clear enough for background removal. Therefore, in order to perform an image segmentation, the wavelength with the greatest separation of reflectance was used to generate the red, green, blue (RGB) images (*.jpg files) using Python. In addition to the RGB images, the grayscale images generated by the Cube Pilot software (Cube-Pilot 1.5.8, Cubert Gmbh, Ulm, Germany) were used as an alpha channel (A), which described the transparency of each pixel, to generate a red–green–blue–alpha (RGBA) image. The additional alpha channel produced more differences between each pixel, resulting in clearer images. The purpose of the RGBA images was solely for segmentation, to be applied as mask images of the original images. The RGBA images were then augmented using rotation, zoom, horizontal and vertical flip. Each set of RGBA images was augmented to a total of 1610 images, where 1127 were used for training and 483 for testing.

### 2.5. Model Development

This section discussed the architecture of the models as well as the workflow for each of the models developed.

#### 2.5.1. Model Architecture

There were three models used in this study, which were developed from the pre-trained VGG16 and Mask RCNN from the Tensorflow model zoo. This section discusses the general architecture of VGG16 and Mask RCNN.

a.    VGG16

The VGG16 model was readily available in Tensorflow and the model was pre-trained with an ImageNet dataset. The architecture of the VGG16 model is shown in Figure 4. VGG16 consisted of five convolutional blocks where each block consisted of convolution layers and a max pooling layer. The five blocks were feature extractors. A dense block was connected to the last convolutional block and the dense layer was a classifier. In this study, the weight of the pre-trained VGG16 model was frozen and was used as a feature extractor and a new dense layer was connected to the convolutional block. The weight of the dense layer was allowed to be trained. This meant that only the pre-trained weight of the convolutional blocks was used and a new classifier was trained with the plant seedling dataset for BSR detection. The model was trained after there was no improvement in the loss for five consecutive epochs. The stopping criterion was set by trial and error. It was

found that after the five consecutive epochs, signs of model overfitting occurred as shown in Figure 5.

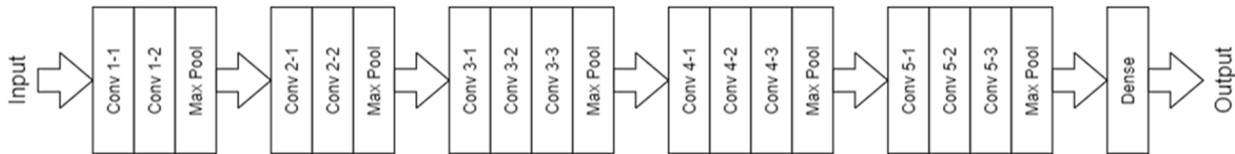

**Figure 4.** The architecture of VGG16.

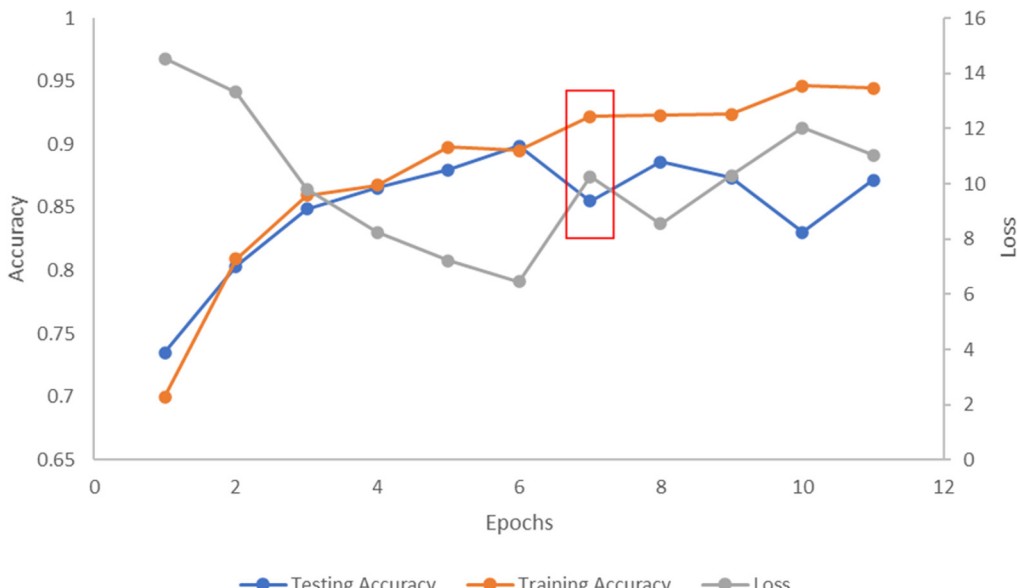

**Figure 5.** Graph of training accuracy, testing accuracy and loss versus epochs. The red box is the epochs after no improvement in the loss for five consecutive epochs.

b.    Mask RCNN

The Mask RCNN is an extension to the Faster RCNN which consisted of two stages. The first stage was a region proposal network (RPN) and the second stage involved class, box offset, and binary mask prediction. The pre-trained Mask RCNN available in the Tensorflow model zoo had a Inception ResNet v2 backbone [41]. The Mask RCNN was an approach for instance segmentation which was based on an instance-first strategy rather than a segmentation-first strategy adopted by other segmentation algorithms [42]. This meant that after the first stage (RPN), in parallel with predicting the class and the box offsets, a mask was also produced. Hence, unlike other algorithms, Mask RCNN does not depend on the mask prediction for classification. Further, most other object detection algorithms utilise RoI pooling for extracting feature maps which compromise the amount of information as quantisation was involved. However, the Mask RCNN used RoIAlign which calculated the value of each sampling points with bilinear interpolation which resulted in the exact value of each sampling point and no quantisation was performed. This allowed the Mask RCNN to predict the mask more accurately. The framework of the Mask RCNN is shown in Figure 6. The Mask RCNN was trained until there was no improvement in the total loss after 3000 consecutive epochs. The stopping criterion was set by trial and error. The epoch with lower loss was taken as the new starting epoch and it was found that by having 3000 consecutive epochs, the lowest total loss could be achieved as shown in Figure 7.

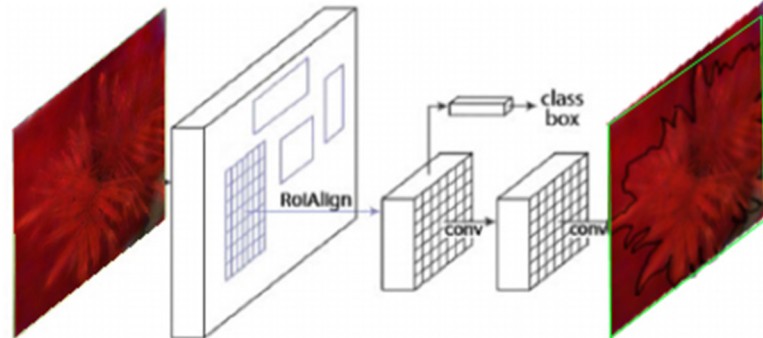

**Figure 6.** The framework of Mask RCNN, which shows that the classification and localisation did not depend on the mask prediction.

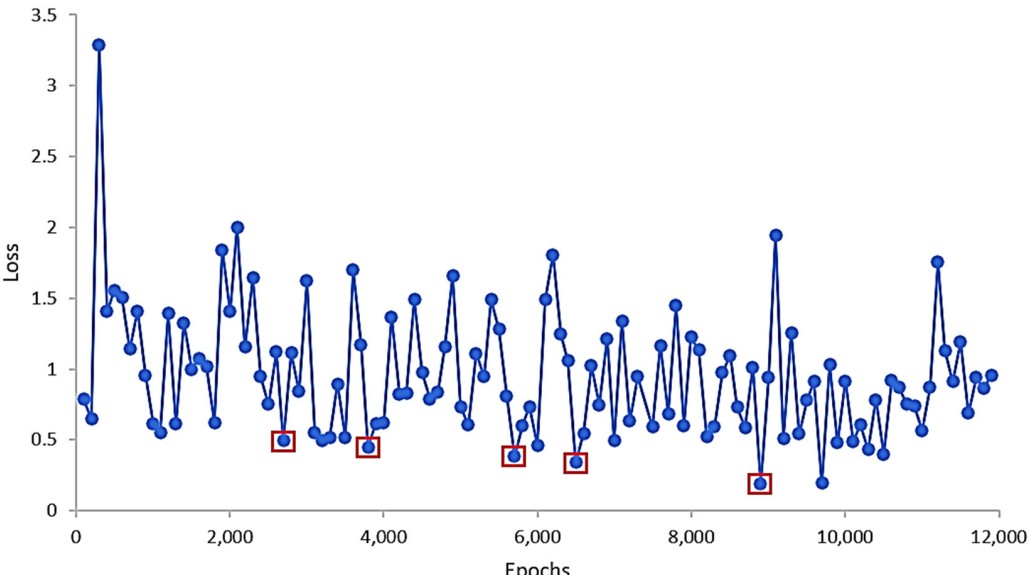

**Figure 7.** Graph of total loss versus epochs. The red boxes are the lowest loss within each 3000 epochs.

2.5.2. Detection Models

Figure 8 shows the workflow of each of the three models used in this study. Descriptions of each model are as follows:

a.      VGG16

The first model used was the VGG16 with the original images generated as in Section 2.4.1 to classify the BSR infected and uninfected seedlings.

b.      Mask RCNN + VGG16

The second model consisted of two stages. The first stage involved creating a mask using the Mask RCNN while the second stage involved classifying the BSR-infected and uninfected seedlings using the VGG 16 trained with the segmented images.

Regarding the first stage of the model, before feeding the images to the Mask RCNN, the RGBA images generated in Section 2.4.2 were digitised and labelled using the "labelme" annotation tool as shown in Figure 9. The images were labelled as frond regardless of the infection condition of the seedling as the purpose of the Mask RCNN was solely to provide the mask images for background removal of the original images. Therefore, the output of the Mask RCNN was a segmented image which consisted of the canopy of the oil palm seedling as the RoI. In the second stage of the model, these segmented images were used to train the VGG16 model to classify the BSR-infected and uninfected seedlings.

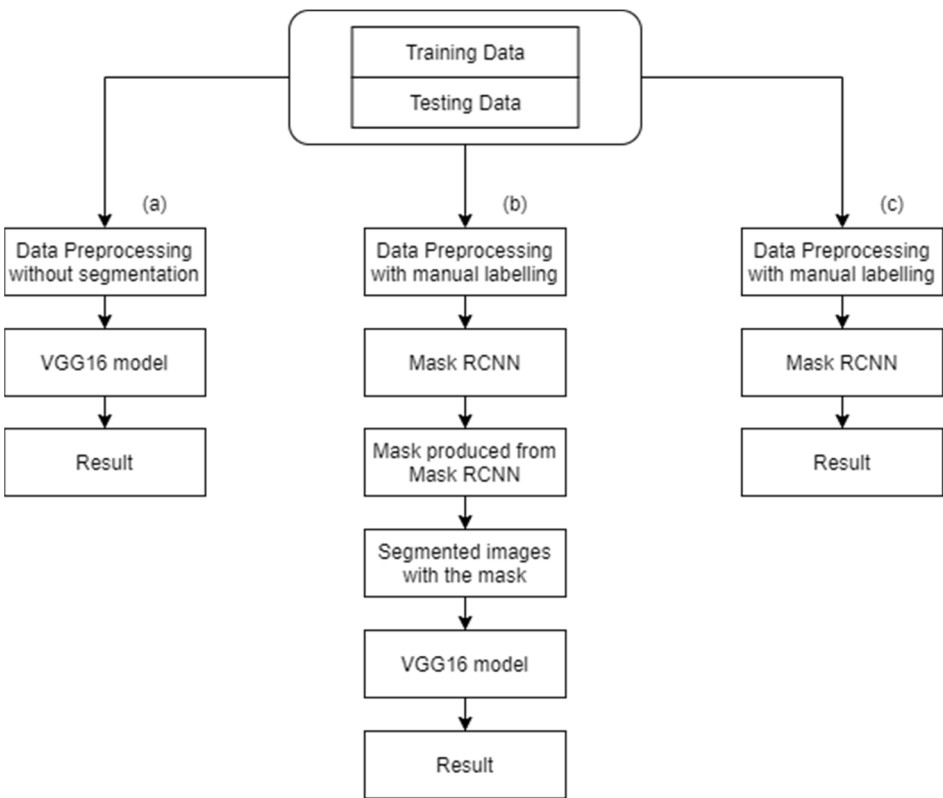

**Figure 8.** Workflow of each classification model. (**a**) VGG16 model trained with unsegmented images, (**b**) VGG16 trained with segmented images produced by Mask RCNN, (**c**) Mask RCNN trained with manually labelled images.

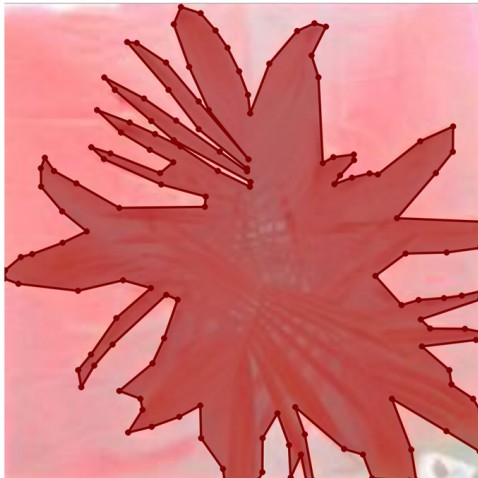

**Figure 9.** Example of the digitised labelled image of an oil palm seedling.

c.    Mask RCNN

The third model emphasised the use of Mask RCNN for object detection, i.e., infected and uninfected seedlings. The original images were digitised and labelled according to the infection conditions with the "labelme" annotation tool. The output of the model was the class of the infection condition, i.e., infected or uninfected.

*2.6. Performance Evaluation of the Models*

The intersection of union (IoU) of the mask produced by the mask RCNN and the manually labelled images were calculated as in Equation (2).

$$IoU = \frac{S_i \cap G_i}{S_i \cup G_i} \tag{2}$$

where $S_i$ is segmented image of image *I* and $G_i$ is the ground truth of image *i*.

The IoU of the segmented images with the ground truth represented the amount of overlap between the segmented images and the ground truth, where the greater the overlap, the greater the IoU.

Meanwhile, the value of accuracy, precision, recall, specificity and F1 score of each model were calculated as in Equations (3)–(7), respectively.

$$Accuracy = \frac{TP + TN}{TP + TN + FP + FN}, \tag{3}$$

$$Precision = \frac{TP}{TP + FP}, \tag{4}$$

$$Recall = \frac{TP}{TP + FN}, \tag{5}$$

$$Specificity = \frac{TN}{TN + FP}, \tag{6}$$

$$F1\ Score = \frac{2 \times Precision \times Sensitivity}{Precision + Sensitivity}, \tag{7}$$

where *TP* is true positive, *TN* is true negative, *FP* is false positive, and *FN* is false negative. The accuracy described the overall correctness of the model for classifying both infected and uninfected seedlings. The precision indicated the percentage of truly infected plant seedlings from the detected infected plant seedlings. The recall, on the other hand, represented the percentage of truly infected plant seedlings detected correctly from all the truly infected plant seedlings. Meanwhile, specificity illustrated the percentage of truly uninfected plant seedlings detected from all the truly uninfected plant seedlings. Besides precision and recall, the F1 score was calculated to obtain the harmonic mean of the precision and recall.

## 3. Results

*3.1. Identified Wavelength for BSR Detection*

Results of the Wilk–Shapiro test for the infected dataset indicated that after performing the log and reflection transformations, some wavelengths demonstrated normality as the data ranges narrowed. Table 1 illustrates the wavelength and data range at which normality was seen in each region after data transformation. After the data were limited to $\mu \pm 0.5\sigma$, the wavelength at 938 nm was found to be normal in all regions. The distribution of selected data before and after transformation in each region at wavelength 938 nm is presented in Table 2. It was demonstrated that the distribution of the data was skewed, and following transformation, the data were fitted to a normal distribution. The data at wavelength 938 nm were analysed using ANOVA and it was shown that there was no significant difference between all the regions at a confidence level of 95% where the *p*-value was 0.1629.

**Table 1.** Wavelengths extracted from infected images that were successfully transformed to a normal distribution after data reduction.

| Dataset | Wavelength (nm) with Normal Distribution after Transformation | | |
|---|---|---|---|
| | RoI = C | RoI = B | RoI = A |
| $\mu \pm 2\sigma$ | None | None | None |
| $\mu \pm 1.5\sigma$ | None | None | None |
| $\mu \pm 1\sigma$ | 910, 914, 918, 922, 926, 930, 934, 938, 942, 946, 950 | 890, 894, 898, 902 | None |
| $\mu \pm 0.5\sigma$ | 906, 934, **938**, 942 | 890, 894, 898, 902, 906, 910, 914, 918, 922, 930, 934, **938**, 942, 946 | **938** |

Note: A: Inner region—2 cm from the centre of the seedling to 5 cm square, B: Middle region—5 cm from the centre of the seedling to 8 cm square, C: Outer region—8 cm from centre of the seedling to 11 cm square.

**Table 2.** Data distribution of the original and transformed data for the RoIs A, B and C at wavelength 938 nm at $\mu \pm 0.5\sigma$.

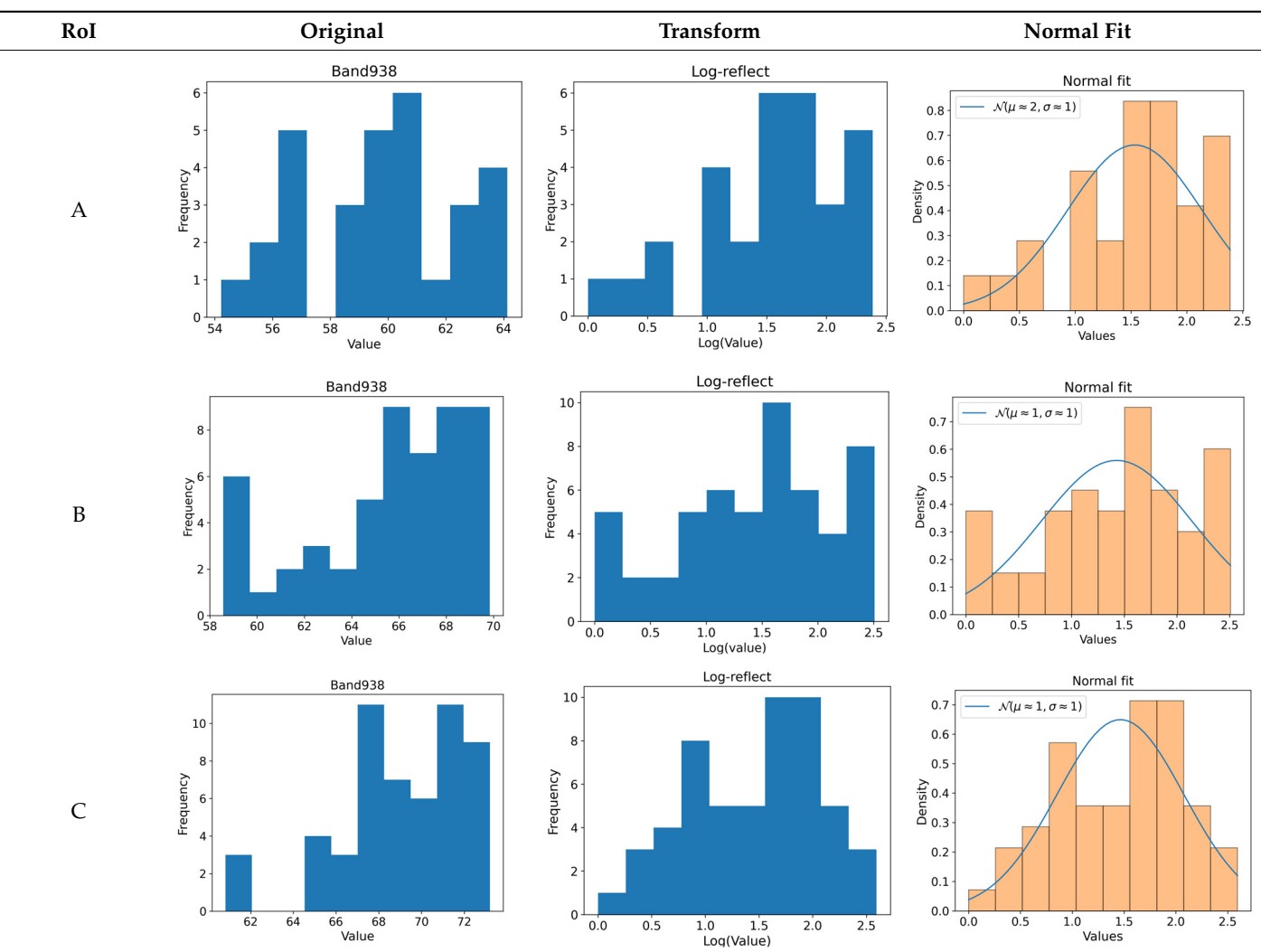

Note: A: Inner region—2 cm from the centre of the seedling to 5 cm square, B: Middle region—5 cm from the centre of the seedling to 8 cm square, C: Outer region—8 cm from centre of the seedling to 11 cm square.

For the uninfected dataset, none of the wavelengths showed normality even after Stage 2 assessment. Therefore, the identified suitable wavelength in the infected dataset, i.e., 938 nm was used to test the significant difference between the data of the three regions

using a Levene's test and a Kruskal–Wallis test. With the results of p-values equal to 0.4234 and 0.3088, respectively, the Levene's test and the Kruskal–Wallis test indicated that there was no significant difference between the three regions. Since there was no effect of leaf geometry on wavelength reflectance at 938nm, the whole canopy images were used to develop the detection models. Further, by reducing the number of bands, the complexity and cost of developing future hardware may be reduced.

### 3.2. Image Segmentation for Background Removal

#### 3.2.1. Identified Wavelengths for Background Removal

Figure 10 shows a graph of the average reflectance of the background (AVG Back), the average reflectance of the plant seedlings (AVG Frond), and the difference between the two (AVG Delta). Reflectance differed most between the red-edge and NIR spectrum, particularly at wavelengths 766 nm, 762 nm, and 770 nm. However, Figure 11a demonstrates that the images created using these wavelengths were insufficiently clear to distinguish between the background and seedling. Therefore, the wavelength with the greatest difference in each red, green and blue spectrum marked in the red box, i.e., at 750 nm (i.e., 10.08% difference), 554 nm (i.e., 3.54% difference), and 466 nm (i.e., 1.37% difference) were chosen. Although the image became better, as seen in Figure 11b, it was not good enough for manual labelling of the contours of the plant seedlings. As illustrated in Figure 11c, when the grayscale image generated automatically by the Cube Pilot software (Cubert Gmbh, Germany) was added into the alpha channel (A), the contrast between canopy and background images became higher. Therefore, the RGBA image was used as the input image of Mask RCNN for the background removal task.

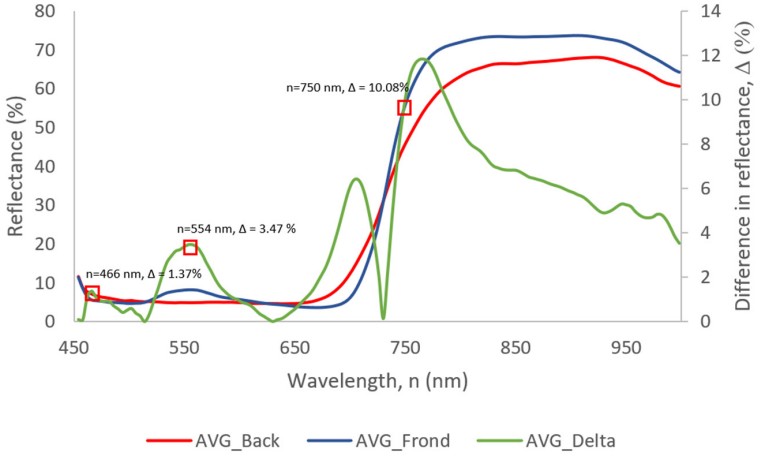

**Figure 10.** Graph of reflectance of the plant seedling and background against bands wavelength.

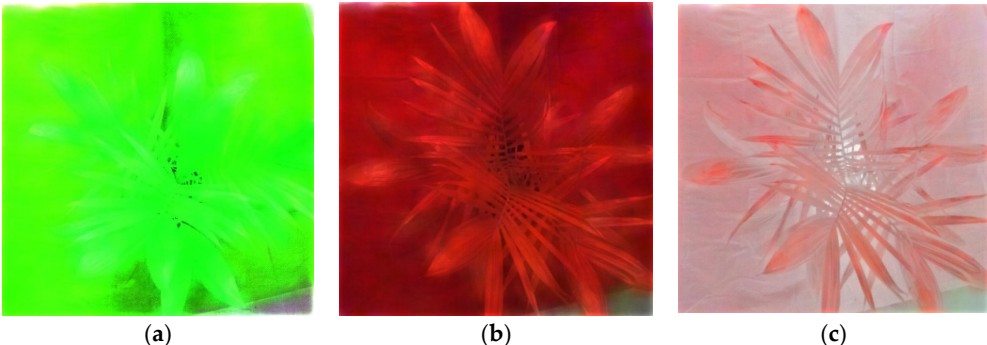

**Figure 11.** Comparison of images generated using (**a**) wavelengths 766 nm, 762 nm and 770 nm and (**b**) images generated using wavelengths 750 nm, 554 nm and 466 nm without the grayscale layer added in the alpha channel and (**c**) with grayscale layer added in the alpha channel.

3.2.2. Performance of the Mask RCNN for Generating Segmented Images

As shown in Figure 12, the final overall loss of the Mask RCNN was 0.2450, with the mask loss at 0.2409 accounting for the majority of the loss. It was demonstrated that the mask (segmented image) rather than the classification of background and foreground was the primary cause of the loss.

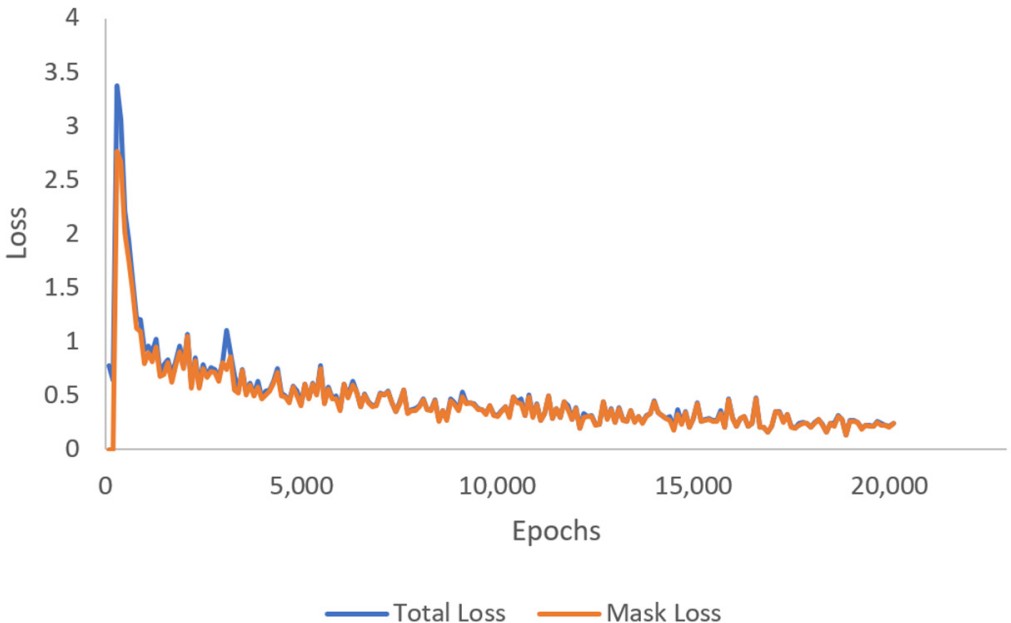

**Figure 12.** Graph of total loss and mask loss versus epochs of Mask RCNN.

The results also demonstrated that the segmented image had an average IoU of 0.8606. A few samples of the segmented image and the ground truth are presented in Figure 13. The IoU of the segmented image was satisfactory in comparison with the other published research. According to [43], the segmentation of a tree crown was deemed proper if the IoU was larger than or equal to 0.5. In addition, [44] employed semantic segmentation for the detection of apple, peach, and pear flowers, with reported IoU values ranging from 0.001 to 0.811 for several deep learning models. Consequently, this demonstrated that the segmented images generated in this study by using the Mask RCNN were acceptable.

In addition to IoU, the values of accuracy, precision, recall, specificity, and F1 score of the Mask RCNN for image segmentation were computed and tabulated as in Table 3, with plant seedling pixel detected as plant seedling pixel as TP, background pixel detected as background pixel as TN, background pixel detected as plant seedling pixel as FP, and plant seedling pixel detected as background as FN. The results demonstrated that 90.48% of the pixels were accurately identified, and 94.63% of the anticipated seedling pixels were in fact seedlings.

**Table 3.** Performance of Mask RCNN for image segmentation.

| Accuracy | Precision | Recall | Specificity | F1 Score |
|----------|-----------|--------|-------------|----------|
| 90.48% | 94.63% | 90.49% | 89.25% | 92.51% |

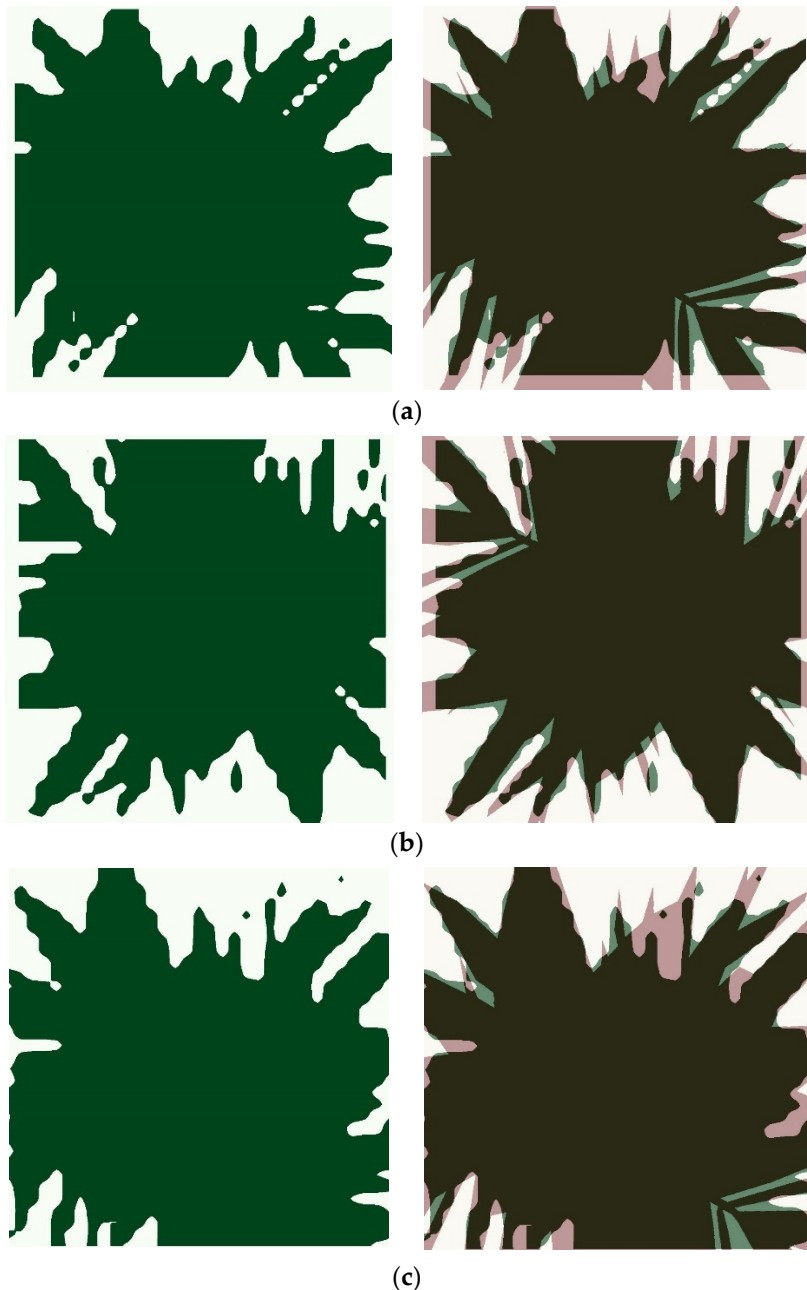

**Figure 13.** Example of the segmented images at different results of IoU. Figures on the left are the mask predicted using the Mask RCNN, while figures on the right are the predicted mask (green) overlayed on the actual mask (red). (**a**) IoU = 0.8598, (**b**) IoU = 0.8829 and (**c**) IoU = 0.8944.

### 3.3. Performance of the BSR Detection Models

Tabel 4 shows the results of the precision, recall, specificity, F1 score, and detection time of the models. It has been demonstrated that the VGG16 model trained on the original images had the highest accuracy (91.93%), precision (94.32%), specificity (94.61%), and F1 score (91.72%). Nevertheless, the VGG16 model trained with segmented images exhibited a better recall (95.02%) than the VGG16 model trained with original images (5.76% difference). The Mask RCNN, on the other hand, had the poorest performance, as only 71.42% of the images were properly identified, but it obtained a 100% recall, as all of the actually diseased seedlings were labelled as such. With a specificity value of 42.74%, Mask RCNN was unable to correctly classify uninfected plant seedlings, as they were classified as infected.

In addition, it was demonstrated that segmentation did not enhance the performance of the model. Despite the limitation of comparable research on the identification of plant diseases, similar findings have been observed in a number of other studies. For example, the authors of [45] reported that the segmentation of skin lesions for dermatoscopic image categorisation reduced the performance of EfficientNet. Furthermore, it was revealed that segmentation-free CNN models performed much better than segmentation-dependent models in the diagnosis of breast masses in mammography datasets [46]. Despite the fact that numerous studies have demonstrated that segmenting the RoI improves model performance, the study only considered Support Vector Machine (SVM) and not CNN [47]. Manual segmentation improved the performance of the SVM model for anatomical magnetic resonance imaging, but no other automatic feature selection approach outperformed the unsegmented image, as demonstrated by the study. In addition, research indicated that feature selection may increase model performance. However, this relies on the employed model and dataset [48]. According to [48], NB, ANN, and a multilayer perceptron did not always obtain improved performance after feature selection, as model performance was dependent on the dataset.

Table 4 also indicates the time required to classify each image. It was discovered that VGG 16 could categorise an image in 0.08 s, but Mask RCNN could only do so in 1.59 s. This was consistent with the difficulty of the task for each model type, as the VGG 16 just classified the images without constructing a mask or bounding box, but the Mask RCNN produced a mask and bounding box for each detected image.

**Table 4.** The performance of the BSR detection models using images extracted from 938 nm wavelength in all regions.

| Model | Segmentation | Accuracy | Precision | Recall | Specificity | F1 Score | Average Time for Classification (s/Image) |
|---|---|---|---|---|---|---|---|
| VGG 16 | No | **91.93 %** | **94.32%** | 89.26% | **94.61%** | **91.72%** | 0.08 |
| VGG 16 | Automatic | 85.46% | 79.79% | 95.02% | 75.93% | 86.74% | 0.08 |
| Mask RCNN | Manual labelling | 71.43% | 63.68% | **100.00%** | 42.74% | 77.81% | 1.59 |

## 4. Conclusions

This paper demonstrates the potential of deep learning to automatically detect an early stage of the BSR disease in oil palm seedlings using NIR-hyperspectral imaging. After data transformation and outlier elimination, it was discovered that the entire structure of an aerial view image of a seedling at 938 nm wavelength may be used for detection, as there are no significant differences in any of the RoI. VGG16 trained with the original images accurately classified BSR-infected plant seedlings with an accuracy of 91.93%. Meanwhile, the Mask RCNN trained using RGBA images correctly segmented the aerial view image of the seedling from the background with an average IoU of 0.8606. This study concluded that the best model for BSR identification is the VGG16 model trained using original images, which allows for more automatic BSR detection as reflectance point extraction is not required. However, this study has limitations due to the controlled environment in which the data was collected. In addition, the data collected only accounted for 10-month-old seedlings which might not represent the unseen data from different growth periods in the nursery. In addition, the other wavelengths that did not exhibit normality were not explored in detail. Therefore, further study can be conducted by putting the model to the test in realistic environments, and the non-normal bands can be further investigated.

**Author Contributions:** Conceptualisation, L.Z.Y. and S.K.-B.; methodology, L.Z.Y., S.K.-B. and F.M.M.; software, L.Z.Y.; validation, L.Z.Y. and S.K.-B.; formal analysis, L.Z.Y. and S.K.-B.; investigation, L.Z.Y. and S.K.-B.; resources, S.K.-B.; data curation, M.J.; writing—original draft preparation,

L.Z.Y.; writing—review and editing, S.K.-B.; visualisation, L.Z.Y.; supervision, S.K.-B., M.J. and F.M.M. All authors have read and agreed to the published version of the manuscript.

**Funding:** This research received no external funding.

**Institutional Review Board Statement:** Not applicable.

**Data Availability Statement:** The data presented in this study are available on request from the corresponding author. The data are not publicly available due to restrictions.

**Conflicts of Interest:** The authors declare no conflict of interest.

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
