# Peer review of "Automatic Disease Detection of Basal Stem Rot Using Deep Learning and Hyperspectral Imaging"

_agriculture, doi:10.3390/agriculture13010069_

Round 1
Reviewer 1 Report
Line 57, please correct link, repeats 22 twice.
Line 57 "terrestrial laser scanning [23–26]," redundant self-citation (researcher Siti Khairunniza-Bejo), one reference is enough.
Table 1-2.4 design needs to be corrected.
Figure 10, it is necessary to sign which charts the axes belong to.
The article states that 938 nm is the most suitable wavelength for detecting the disease, while the results in Table 4 are given for 934 nm.
Why do the authors propose to use only one wavelength for the analysis to detect the disease? The use of images with multiple channels usually improves the accuracy of detecting the desired features, especially since hyperspectral images were analyzed in this study.
Reviewer 2 Report
Reconsider after major revision

Reviewer 3 Report
This manuscript touches upon the actual problem of diagnosing plant diseases.
There are the following questions and comments on the text of the Manuscript:
1. Line 38: Convert the amount of losses to USD.
2. Lines 58-60: why, according to [28], HSI methods have great potential in comparison with the rest?
3. What kind of illumination was used when obtaining hyperspectral images?
4. Why was the hyperspectral camera exactly at a height of 2.6 m?
5. There are extraneous characters in lines 211-212, 222, 242, 246, 266.
6. Figures 5, 7 and others should be improved: remove the frames, increase the contrast.
7. It is necessary to improve the quality of the figures in Table 2: very small font.
8. Section 3.2.1 does not sufficiently convincingly show the choice of wavelength. There are no numerical criteria.
